# Synthetic Lethality in Lung Cancer—From the Perspective of Cancer Genomics

**DOI:** 10.3390/medicines6010038

**Published:** 2019-03-12

**Authors:** Iwao Shimomura, Yusuke Yamamoto, Takahiro Ochiya

**Affiliations:** 1Division of Molecular and Cellular Medicine, National Cancer Center Research Institute, 5-1-1 Tsukiji, Chuo-ku, Tokyo 104-0045, Japan; ishimomu@ncc.go.jp (I.S.); yuyamamo@ncc.go.jp (Y.Y.); 2Department of Respirology, Graduate School of Medicine, Chiba University, 1-8-1 Inohana, Chuo-ku Chiba-shi, Chiba 260-8670, Japan; 3Institute of Medical Science, Tokyo Medical University, 6-1-1 Shinjuku, Shinjuku-ku, Tokyo 160-8402, Japan

**Keywords:** lung cancer, cancer genome, synthetic lethality

## Abstract

Cancer is a genetic disease, and this concept is now widely exploited by both scientists and clinicians to develop new genotype-selective anticancer therapeutics. Although the quest of cancer genomics is in its dawn, recognition of the widespread applicability of genetic interactions with biological processes of tumorigenesis is propelling research throughout academic fields. Lung cancer is the most common cause of cancer death worldwide, with an estimated 1.6 million deaths each year. Despite the development of targeted therapies that inhibit oncogenic mutations of lung cancer cases, continued research into new therapeutic approaches is required for untreatable lung cancer patients, and the development of therapeutic modalities has proven elusive. The “synthetic lethal” approach holds the promise of delivering a therapeutic regimen that preferentially targets malignant cells while sparing normal cells. We highlight the potential challenges in synthetic lethal anticancer therapeutics that target untreatable genetic alterations in lung cancer. We also discuss both challenges and opportunities regarding the application of new synthetic lethal interactions in lung cancer.

## 1. Introduction

The concept of synthetic lethality was introduced almost 100 years ago, and it has recently been exploited in various efforts to develop new genotype-selective anticancer therapeutics [1]. Synthetic lethality is defined as a condition in which a simultaneous mutation in two genes (‘A’ and ‘B’) leads to cell death but the mutation of either gene alone is compatible with viability [2,3] (Figure 1). In other cases, alteration of one gene may cause a damaging phenotype, but the system compensates through genetic buffering to maintain the health of organism; however, the occurrence of an alteration to the second gene would be lethal when it occurred. Targeting synthetic lethal vulnerabilities in cancer can cause cancer cell death without harming normal cells because cells harboring cancer-related alterations should have different sensitivities than cells without cancer-related alterations. Thus, synthetic lethality is one of the hopeful approaches for future clinical applications.

The search for synthetic lethal interactions was first studied in fruit flies and expanded to other model systems, including yeast and *Caenorhabditis elegans* (*C. elegans*) [4,5]. Mutations in cancer genes may be classified into loss-of-function or gain-of-function defects; synthetic lethality interactions are more commonly associated with loss-of-function alleles but can also apply to gain-of-function alleles. The synthetic lethality concept has been expanded to include a number of other related genetic concepts, such as induced essentiality, collateral sensitivity and synthetic dosage lethality, which are termed for a more recent concept of synthetic lethality [6]. To date, the most characterized synthetic lethal interactions described are breast cancer susceptibility genes 1 and 2 (BRCA1 and BRCA2) loss-of-function with inhibition of poly (adenosine diphosphate [ADP]-ribose) polymerase (PARP), resulting in drastically sensitized cancer cells [7,8]. Furthermore, studies targeting undruggable oncogenes such as RAS and MYC can be approached by synthetic lethal interactions [9,10,11].

With a more complete understanding of the complex and extensive network of untreatable target effectors and regulators, various therapeutic approaches have been developed. The secondary dependencies on genes that have been exploited themselves are not oncogenes but could lead to vulnerabilities caused by another mutation state [12,13]. Nonetheless, it remains unclear to what extent synthetic lethal interactions proved in experiments can be applied to clinical settings because multiple factors, such as epigenetic and tumor micro-environmental factors, have been postulated to cause effects [14]. During the process of cancer evolution, cells acquire “hallmarks of cancer,” which are not responsible for initiating tumorigenesis, but they are common characteristics of many cancers. Non-oncogene addiction involves genes and pathways that are essential to support the oncogenic status of cancer cells but are not mandatory to the same degree for normal cell growth and is related to gene interactions in the cancer atmosphere [15]. Potentially, any of these factors might emerge as a treatment target. However, its statistical power is limited by the mutation frequency. For example, assuming that the frequency of an expected co-mutation of two genes is 1%, the possibility of occurrence would be 0.01%, which means that 10,000 samples are required for a single assay. To solve this issue, the construction of experimental models and the development of highly specific technological innovation have been promoted.

Approaches to synthetic lethal screens have now progressed from models to small-molecule inhibitor screening and genome-wide small interfering RNA (siRNA) screening to develop potential new treatment modalities [12,16]. Moreover, major advancements in high-throughput screening platforms and the development of CRISPR-Cas9 technology have further promoted the capacity to uncover synthetic lethal interactions in cancer cells [17,18] (Figure 2). The emergence of powerful genetic tools has revealed correlations between chemotherapeutic responses and specific genetic backgrounds in lung cancer. We provide an overview of the recent progress made in a synthetic lethal approach to lung cancer, including the most promising strategies for future developments.

## 2. From Cancer Genome to a Synthetic Lethal Approach

It is now clear that cancer is a genetic disease caused by changes in, or damage to, one or more genes [19,20]. To understand the complex differences between cancer cells and normal cells, gene expression was evaluated in various types of tumors over the decade. A single cancer cell normally has mutations in multiple genes, DNA rearrangements, and gross chromosomal abnormalities, leading to widespread alterations in its gene expression profile. When a gene has some defect, such as missing or overactivation, the change in the DNA sequence of the genome can occur and leads to cancer [21,22].

Early insights into the central role of the genome in cancer development were observed in the late 19th and early 20th centuries [23], and the discovery of DNA and the determination of its structure led to the concept that DNA damage and generations of mutations can cause cancer [24,25,26]. Researchers have established that the multistage process of tumor formation is driven by activating mutations in dominant growth-enhancing genes (oncogenes) and inactivating mutations in recessive growth-inhibitory genes (tumor suppressor genes) [27]. As is well known, mutations have multiple effects on cell proliferation but also in cell fate determination, such as differentiation, apoptosis, and senescence. The phenomena of oncogene addiction and tumor suppressor gene hypersensitivity suggest that the multistage process of carcinogenesis, consistent with the fact that cancer-related proteins affect interacting networks, is not simply a summation of the isolated effects of oncogene activation and tumor suppressor gene inactivation [28]. The research on oncogenes and tumor suppressor genes accelerated the quest for considerable complexity in the mutational origins of cancer with multiple genes contributing to carcinogenesis [29]. In 1986, Renato Dulbecco mentioned that the complete sequence of the human genome would be an essential tool for systematically discovering the genes that result in carcinogenesis [30]. The Human Genome Project (HGP) was launched in 1990 and completed in 2003, and it greatly impacted the ability to investigate biological phenomena in a comprehensive, unbiased, hypothesis-free manner [31,32,33].

The HGP, nominally completed in 2004, aimed to sequence the human genome [34]. The genome-wide analysis has provided substantial information for understanding genes and their mutations. The sequences have allowed the systematic study of the germline status of encoded genes and have led to the discovery of somatic mutations in cancer cells. There is a general consensus for cancer genes; mutations in more than 1% of genes contribute to human cancer [35]. These platforms of human genomes have led to new therapeutic strategies for precision medicine. The large-scale, systematic sequencing studies conducted so far in the past decades have provided insights into the mutational processes operating in human cancer; new observations have served both to clarify and to modify understanding of the process of tumorigenesis and metastasis. Hanahan and Weinberg proposed six hallmarks of cancer that provide a logical framework for understanding the diversity of neoplastic diseases: genome instability, unlimited cell division, sustained proliferative signaling, evasion of growth suppression, cellular energetics and apoptosis resistance [29,36]. However, numerous questions have remained regarding how mutations in DNA lead to the acquisition of these phenotypes and traits. Genetic changes are likely to arise from endogenous reactions of chemical substances with DNA and physiological processes but also from the effects of exogenous mutagenic factors, such as tobacco smoke, ultraviolet light, and even from trivial elements in the atmosphere. Clearly, complex cellular systems do not arise from the independent action of the function of genes themselves but, rather, the result of broad genetic interactions among each gene.

Beyond the discovery of somatic mutations of cancer genes affecting global processes in the cancer genome, insights into the signaling circuitry that facilitates intercommunication between the cells involved in tumorigenesis will surely accelerate the efforts to develop a resource for cancer genotype-to-phenotype relationships. A genome-scale genetic interaction map was constructed by examining millions of gene-gene pairs, and a network based on genetic interaction profiles revealed a functional map of a cell in which genes of similar biological processes cluster together [37]. Genetic interactions can also determine the relationship between genotype and phenotype in current genome-wide association studies [17]. Large-scale genetic interactions can contribute to the identification of a synthetic lethal therapeutic approach, which is of particular interest for harnessing new cancer treatment targets. With further improvement of computer processing and with lowering sequencing costs, there is the possibility for whole genome sequencing at the bedside. Even with the remarkable advances in sequencing technology, we are now in the antediluvian era before the cancer genomics flood. Advances in the technological next-generation sequencing of circulating, tumor-derived nucleic acids hold promise for addressing the challenge of developing safe and effective sequencing for research and individual cancer-patient genomes [38].

## 3. Therapeutic Advances in Lung Cancer

In the past, most chemotherapeutic agents were developed as monotherapy for various kinds of tumors. From the viewpoint of lung cancer, close to 70% of patients present with locally advanced or metastatic disease on arrival, and the combination of one or more of these drugs is beneficial for these patients. Various clinical trials were organized, and cisplatin-based combined chemotherapy for metastatic non-small cell lung cancer (NSCLC) and small cell lung carcinoma (SCLC) resulted in a small but statistically significant improvement in overall survival compared with the best supportive care. Although these therapeutic approaches with cytotoxic agents were adequate for many patients with lung cancer, the treatment effects have plateaued. The treatment of lung cancer, also for other cancer types, has changed from cytotoxic chemotherapy to more personalized medicine.

Carcinogenesis has long been recognized as a state of uncontrolled cell growth; the discovery of the structure of DNA led to the initiation of determining genetic alterations involved in carcinogenesis. It is now well appreciated that tumors require an increase in genetic disorder and epigenetic instability, including the downregulation of mechanisms related to genetic damage tolerance. We have to correspond to these mechanisms that they maintain uncontrolled cell progression and metastatic ability that is associated with heinous cancer cells. After the advance of cytotoxic agents, the era of molecular targeted drugs has come, and whole genome sequencing offered by next-generation sequencing provides target genes. The first clinical success for molecular targeted chemotherapy was imatinib, a tyrosine kinase inhibitor for chronic myeloid leukemia (CML), characterized by the presence of the Bcr-Abl fusion gene [25,39]. The identification of gene mutations is becoming more important, and it helps to make decisions for patients. For example, tyrosine kinase inhibitors (gefitinib, crizotinib) were developed for NSCLC [40,41], a monoclonal antibody (trastuzumab) was developed for breast cancer [42], and recently, immunotherapy, such as anti-cytotoxic T lymphocyte-associated protein 4 (CTLA-4) antibodies (ipilimumab) and anti-programmed cell death protein 1 (PD1) antibodies (nivolumab) [43,44], has received significant attention. These trends suggest that we are now entering the era of precision medicine that is based on the driver genes of cancer cells. In lung cancer, there is the rapid development of targeted therapies with newer and more potent generations of drugs; cancer cells have acquired mechanisms of resistance, particularly in patients with EGFR mutations, in which the most common resistant cause, the T790M mutation, has been successfully treated with osimertinib [45].

It is certain that major progress has been made through molecular targeted therapy; however, as many as half of the solid tumors may not harbor a known oncogene, and some genes, such as KRAS, have no established therapeutic methods. Another approach has been proposed: synthetic lethality. Over the past decades, there has been enormous progress in the understanding of the biology and mechanisms of cancer cells. More than the use of new therapeutic modalities, a further approach for overcoming the struggle with cancer cells is warranted. In the following paragraphs, we will describe illustrative examples of synthetic lethality interactions of diverse cellular pathways in lung cancer with an emphasis on those that might be clinically acceptable. In addition, technological innovations and historical challenges that might shape the discovery and development of new synthetic lethal approaches are described.

## 4. Discovering Synthetic Lethal Interaction in Lung Cancer

Lung cancer is the leading cause of cancer death worldwide, and it is estimated that more than 1 million deaths occur per year [46]. NSCLC is the main group of histological subtypes of lung cancer: adenocarcinoma and squamous cell carcinoma are the most common subtypes, accounting for approximately 85% of all lung cancers; SCLC accounts for 15% [47,48]. In advanced lung cancer, cytotoxic chemotherapy has improved the prognosis of NSCLC, and new advances in the discovery of oncogenic drivers and specific targeted therapies have brought significant improvement in the outcomes and quality of life of NSCLC patients [49]. The most commonly mutated genes in lung adenocarcinoma are KRAS, EGFR, ALK, TP53, STK11, KEAP1, CDKN2A and SMARCA4 [50]. There are various other related gene mutations, including those that are not currently clinically actionable ones. Highlighting previously unappreciated altered genes may provide further refinement for treatable approaches. Although most targeted genes developed for lung adenocarcinoma are mainly ineffective against lung squamous cell carcinoma, commonly mutated genes in lung squamous cell carcinoma are TP53, NFE2L2, KEAP1 and CDKN2A [51]. Previous studies have identified these potential target genes, and this data will help to design appropriate clinical trials to analyze the efficacy of targeted therapy in lung squamous cell carcinoma. In SCLC, targeted therapy is not well established, and almost all cases are linked to inactivating TP53 and RB1 mutations [52]. The identification of targetable gene alterations has transformed the management of lung cancer; however, some intracellular oncogene products have proven difficult to target.

### 4.1. Synthetic Lethality in NSCLC

Approximately 85% of patients can be grouped by histological subtypes of NSCLC, of which lung adenocarcinoma (LUAD) and lung squamous cell carcinoma (LUSC) are the most common subtypes [47]. Major subtypes of NSCLC are associated with smoking; the identification of driver gene alterations has allowed targeted therapies. Progress in this area has brought substantial and promising results in patients who have specific predictive biomarkers and received targeted therapy or immunotherapy compared with those who received chemotherapy [49]. Although crucial therapeutic developments have been established for patients with NSCLC, there are still major challenges to be elucidated. New strategies for untreatable target genes, better understanding of overcoming resistance to molecular targeted therapy, and the identification of new biomarkers to provide precise immunotherapy are indispensable. Thus, much progress has been made in research, and new therapeutic approaches, such as synthetic lethality, have emerged to establish novel therapies for NSCLC (Table 1). Genetic screening is the most mainstream method currently, and also chemical screening is widely accepted that determines associations between mutated oncogenes and specific compounds. Through a chemical library screen, oncrasin-1 (oncogenic Ras tumor-inhibiting compound 1) showed synthetic lethality for KRAS and protein kinase C iota (PKCi) [53], and phenformin, a mitochondrial inhibitor and analog of the diabetes therapeutic metformin, induced apoptosis in LKB1-deficient NSCLC cells [54]. KRAS and PKCi are in so-called gain-of-function/gain-of-function associations, and phenformin treatment in LKB1-deficient NSCLC cells has loss-of-function/loss-of-function associations. Although the terminology of synthetic lethality is traditionally associated with loss-of-function/loss-of-function associations, there is a necessity to further search for gain-of-function treatments that could lead to synthetic lethality in combination with another gain-of-function treatment. While the synthetic lethal interaction between BRCA and PARP inhibition was predicted from the DNA repair mechanism, large-scale RNA interference (RNAi) screening has emerged. RNAi-based functional genomics provides the opportunity to derive a comprehensive analysis of validated gene targets that support synthetic lethality research. First, high-throughput cell-based screening with an siRNA library revealed several sensitizers for lung cancer cells to paclitaxel [55]. Thereafter, a large number of RNA interference-based synthetic lethality screenings were promoted, especially those targeting RAS mutations. As RAS signaling pathways are essential for the normal development of cells and homeostasis of the internal environment, it will be difficult to achieve a good therapeutic effect in many cases. The expression of oncogenic RAS in normal cells often results in apoptosis or senescence, whereas in immortal cells or in cells with inactivation of p53, the expression of oncogenic RAS results in transformation and carcinogenesis, which means that RAS will be the target of synthetic lethality even in normal cells. The unsurpassable obstacle of RAS led to an attempt to explore the interactions of synthetic lethality between the expression of mutated RAS oncogenes and other loss-of-function genes in association with gain-of-function/loss-of-function genes. The main targeted genes in activated mutant RAS are STK33, TBK1, PLK1, WT1, CDK4 and GATA2 as synthetic lethal targets [12,56,57,58,59,60]. Similar studies have been conducted for other cancer types, such as colon cancer, and their relationships and differences are also noted. Other combinations of genes were also discovered, such as SMARCA2 and BRG1 [61]. Although the RNAi screening method has brought robust development in searching for genetic-based synthetic lethality, CRISPR-Cas9 screening technology as an alternative means of a genetic approach has enabled a platform to uncover redundant genes and to explore complex gene networks. Despite the initial success of RNAi screening, the technology has been hampered by problems with off-target effects and effect reproducibility. The use of the CRISPR-Cas9 genome screening system enabled more complete loss-of-function effects by targeting DNA and identifying essential genes, rather than disrupting mRNA. In a study using a database of human acute myeloid leukemia (AML) cell lines, a genome-wide CRISPR-based screen revealed PREX1 as an AML-specific activator of MAPK signaling in RAS processing [62]. The synthetic lethality of the von Hippel-Lindau (VHL) tumor suppressor gene and HIF activity correlated with histone methyltransferase EZH1 function were reported in renal cancer with CRISPR-Cas9 system elimination [63]. As various approaches have been utilized in other cancer types, it is desirable to further promote studies using the latest technology, including CRISPR-Cas9 and other modalities.

### 4.2. Synthetic Lethality in SCLC

SCLC accounts for approximately 15% of all lung cancers, and the tumor cells express neuroendocrine markers [71]. Although 95% of small cell carcinomas originate in the lung, they can also arise from extrapulmonary sites, including the nasopharynx, gastrointestinal tract, genitourinary tract and other sites [72]. SCLC frequently responds to chemotherapy; recurrence arises rapidly in the majority of cases, and the prognosis is quite poor because surgery is rarely fully successful [73]. There are few improvements in the fundamental treatments for SCLC, with most advances being brought by improvements in radiotherapy approaches. Although targeted therapies have successfully achieved therapeutic responses in NSCLC, advances have been lacking in SCLC [74]. Determining complex genomic alterations and rearrangements in SCLC, which are undetectable by the existing gene analysis technologies, might further contribute to understanding of SCLC occurrence; therefore, performing whole-genome sequencing is necessary; one of the hallmarks of SCLC is the high frequency of mutations in TP53 and RB1 [75,76]. Most of the discovered mutations observed in SCLC are passengers that do not essentially contribute to cancer cell growth, invasion or progression to metastatic lesions, and the most common mutations, that is, those in TP53 and RB1, are rarely targeted directly. Unlike adenocarcinoma, gene mutations in SCLC are not broadly characterized by driver mutations such as tyrosine kinase activation. Approaches from other aspects, such as transcriptional regulation, histone modification and cytoskeleton regulation, are not yet options for treatment; they should be expanded further, and such an effort may provide functionally tractable information. The comprehensive analysis of somatic alterations in SCLC provides proof for universal bi-allelic inactivation of TP53 and RB1. The genomic analyses also identified NOTCH family genes as tumor suppressors and regulators of neuroendocrine differentiation in SCLC [52]. Unlike NSCLC and other subtypes of lung cancer, there has been no breakthrough for SCLC treatment for more than 25 years, and the only agent approved is topotecan for second-line therapy. Most of patients receive a platinum doublet combination therapy of etoposide; although it is initially effective, recurrence arises rapidly in the vast majority of cases. The inability to fully destroy residual cells in SCLC, despite its drastic initial sensitivity to chemotherapy, suggests that SCLC cells possess cancer stemness that is relatively resistant to cytotoxic therapy [48]. The potential benefits of gene-targeted therapy have been studied for SCLC, and recent work showing the frequent occurrence of genomic TP73 alterations in SCLC indicates that there are potentially promising targets in SCLC tumors [52]. The benefits of targeted therapy for other newly discovered genes, such as KIAA1211, COL22A1, ASPM, PDE4DIP or PTGFRN, remain unknown. For the difficulties of targeting a single gene, recent studies have focused on a synthetic lethality interaction approach to provide a paradigm for targeting SCLC cells in which a loss of tumor suppressor function has occurred (Table 1). Large-scale RNAi screening has uncovered PRKDC for synthetic lethality in cells overexpressing MYC; by modulating mRNA and protein expression levels DNA double-strand breaks were induced [65]. Tumor-specific inactivation of the MYC-associated factor X gene (MAX) with alterations of BRG1 revealed synthetic lethality related to an aberrant SWI/SNF-MYC network in SCLC [66]. Both the RB1 and CDKN2A genes are tightly associated with cell cycle regulation, and knockdown of the RB1 gene in CDKN2A-mutant cells resulted in inhibition of cell proliferation [67]. MAX/BRG1 and RB1/CDKN2A both arose from computational analysis and are characteristic loss-of-function tumor suppressor genes. Treatment with JQ1, a BET inhibitor, and shRNA library screening identified HDAC6 as a synthetic lethal target in an NK cell-mediated manner [68]. Recent developments in CRISPR-Cas9 screening has revealed the loss-of-function of the RB1 tumor suppressor, heralding a synthetic lethal interaction with Aurora A and B kinase inhibition [69,70]. In the process of cancer progression, loss-of-function mutations are common events, and targeting these mutations via synthetic lethal interactions will lead to an expansion of therapeutic approaches toward tumors without dominant oncogenes, such as SCLC.

## 5. Future Prospective

Synthetic lethal interactions for lung cancer have provided numerous opportunities for novel therapeutic approaches, and technological advances will provide a more expanded context of synthetic lethal interactions. Further capabilities for genetic mapping in human cancer cells, far beyond the identification of newly discovered genes, require extensive validation and confirmation to rigorously demonstrate the value of those synthetic lethal interactions. Screening using small molecules and using genetic engineering techniques such as RNAi and CRISPR-Cas9 are progressing through efforts to further explore synthetic lethality. Targeting oncogenes with chemical inhibitors or treatment with specific antibodies has proven to exert effects for cancer therapy; however, the function of tumor suppressor genes in clinical stages has not been elucidated. A new synthetic lethality concept, rather than targeting tumor suppressor genes themselves, has emerged and is obtaining achievements in SCLC fields. Even when comparing adenocarcinoma and squamous cell carcinoma, the expression status of oncogenes and tumor suppressor genes and the relationship of that status with other pathways are heterogeneous; therefore, a novel approach from another perspective is required. While vast analyses over the past decade regarding synthetic lethality in lung cancer have focused on RNAi approaches, CRISPR-Cas9-based approaches will provide increased and superior throughputs, with effort at a lower cost. In addition, for the CRISPR system, a novel approach of genome-wide libraries for CRISPR-based gene inhibition (CRISPRi) and activation (CRISPRa) is emerging to efficiently determine gene functions with multiple modalities [77]. Studies using RNAi and CRISPR-Cas9 have led to more specific research in gene editing, but in recent years, these studies have mainly used cell lines, and their evaluation of how cancer cell heterogeneity relates to therapeutic interventions is still insufficient. Given the genetic and epigenetic heterogeneity of tumorigenesis, combinatorial therapeutic approaches will most likely be required. From the viewpoint of tumor heterogeneity, artificial situations such as clear knockdown or upregulation of single or specific genes do not necessarily maintain the tumor characteristics seen in clinical settings. New insights through new technologies such as CRISPR-Cas9, primary culture systems, and organoid culture systems are needed; the fusion of various experiments for translating results into clinical practice is warranted.

## Figures and Tables

**Figure 1 medicines-06-00038-f001:**
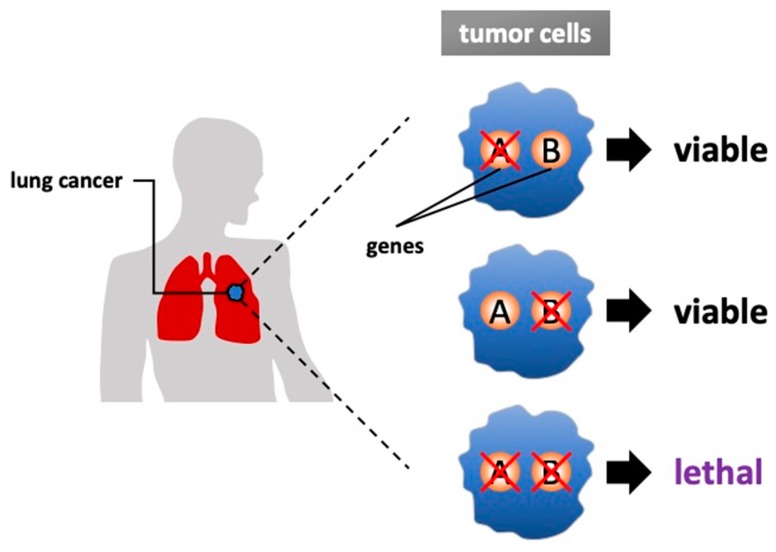
Synthetic lethality in lung cancer. Two genes are defined as synthetically lethal if simultaneous mutations of both genes cause cell death while sparing normal cells. Most cancers harbor one or more gene mutations that are categorized as loss-of-function or gain-of-function, and synthetic lethality is targeting these combinations. Basically, a genetics-based approach and a drug-based approach are used for the research to identify the targetable genes.

**Figure 2 medicines-06-00038-f002:**
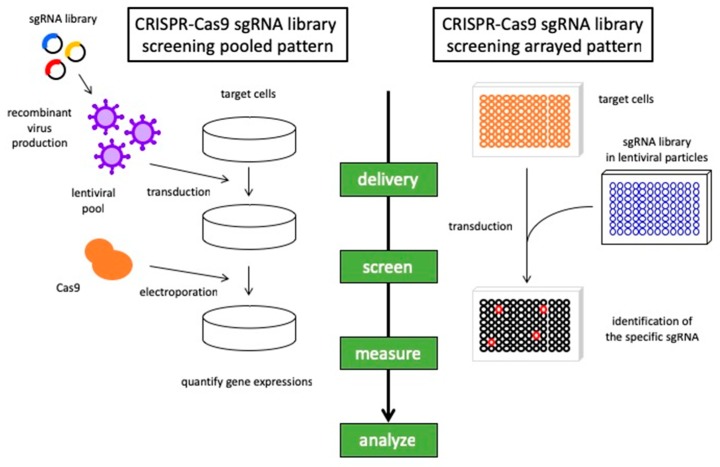
Synthetic lethality screening with CRISPR libraries. The general workflow for synthetic lethality screening using the CRISPR-Cas9 system in pooled and arrayed assay methods. In a pooled pattern, transducted cells are subjected to the next-generation sequencer to provide the enrichment of gene expression by the selection of sgRNA. In an arrayed pattern, cells are transducted by lentivirus containing sgRNA libraries and the phenotypes were evaluated corresponding to the sgRNA. The basic principle is common, and it is desirable to choose the method according to the situation.

**Table 1 medicines-06-00038-t001:** Synthetic lethal combination and approaching method in lung cancer.

Category	Genomic Alteration (Indicator 1)	Synthetic Target (Indicator 2)	Types of Lung Cancer	Approaching Method	References
oncogene	KRAS	protein kinase C iota (PKCi)	NSCLC	chemical library screen	[53]
oncogene	KRAS	PLK1	NSCLC	RNA interference screen	[12]
oncogene	KRAS	STK33	NSCLC	RNA interference screen	[57]
oncogene	KRAS	TBK1	NSCLC	RNA interference screen	[58]
oncogene	KRAS	WT1	NSCLC	RNA interference screen	[59]
oncogene	KRAS	CDK4	NSCLC	RNA interference screen	[60]
oncogene	KRAS	GATA2	NSCLC	RNA interference screen	[56]
other signaling	LKB1	phenformin	NSCLC	chemical library screen	[54]
chromatin remodeling	BRG1	SMARCA2	NSCLC	RNA interference screen	[61]
oncogene	KRAS	EGFR	NSCLC	computational analysis	[64]
transcription factor	MYC	PRKDC	SCLC	RNA interference screen	[65]
transcription factor	MAX	BRG1	SCLC	computational analysis	[66]
tumor suppressor gene	RB1	CDKN2A	SCLC	computational analysis	[67]
transcription factor	BET	HDAC6	SCLC	RNA interference screen	[68]
tumor suppressor gene	RB1	Aurora A	SCLC	CRISPR screen	[69]
tumor suppressor gene	RB1	Aurora B	SCLC	CRISPR screen	[70]

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
