# Peer review of "Synthetic Lethality in Lung Cancer—From the Perspective of Cancer Genomics"

_medicines, 2019, doi:10.3390/medicines6010038_

Reviewer 1 Report

So far there are many excellent reviews on the topics of synthetic lethal. This review focuses on lung cancer, and provided some update information on the related research advances in lung cancer, and has some value for readers in the field. However, the writing needs extensive improvement.  

The writing needs extensive improvement . Here I just give couple of examples:

 programmatic errors:

“When a gene has some defect, is missing, or is overactivated, the change in the DNA sequence of the genome can occur, leading to cancer”

“The phenomena of oncogene addiction and tumor suppressor gene hypersensitivity suggest that the multistage process of carcinogenesis, consistent with the fact that products of these processes, proteins that affect interacting networks,is not simply a summation of the isolated effects of oncogene activation and tumor suppressor gene inactivation” .   The reader will be confused by the words underlined.

“There is a general consensus for cancer genes thatindicates that mutations in more than 1% of genes contribute to human cancer” 

Inappropriate expression:

“As is well known, mutations play an important role in cell proliferation but also in cell fate determination, such as differentiation”. 

Mutation has multiple effects on cell proliferation, or differentiation, or cell fate determination, but one should not say mutations play important role in those processes. 

“These studies accelerated the quest for considerable complexity in the mutational origins of cancer, with cancer- causing genes varying across and within tumor types and with multiple genes contributing to carcinogenesis”

What does “These studies” refer to in the context of this paragraph?

 Some sentence should not appear on such a professional review. For example:

 “Genes are coded messages that produce proteins, and the proteins control the way cells behave.”

“The HGP, nominally completed in 2004, aimed to sequence the human genome [34]. The

genome-wide analysis has provided substantial information for understanding genes and their mutations; reports of genome sequences ripple beyond biomedical science and technology into the social, economic, and political spheres.” 

Author Response

Response to Reviewer 1 Comments

We are grateful to the reviewer for their critical comments and suggestions that have helped us to improve our manuscripts. As indicated below, we have modified all of these suggested programmatic errors and inappropriate expressions.

Reviewer 1:

The writing needs extensive improvement. Here I just give couple of examples:

programmatic errors:

Point 1: “When a gene has some defect, is missing, or is overactivated, the change in the DNA sequence of the genome can occur, leading to cancer”

Response 1: Thank you for your suggestion. We have revised the writing as “When a gene has some defect, such as missing or overactivation, the change in the DNA sequence of the genome can occur and leads to cancer”.

Point 2:“The phenomena of oncogene addiction and tumor suppressor gene hypersensitivity suggest that the multistage process of carcinogenesis, consistent with the fact that products of these processes, proteins that affect interacting networks, is not simply a summation of the isolated effects of oncogene activation and tumor suppressor gene inactivation” .   The reader will be confused by the words underlined.

Response 2Sorry for confusing writing, and to avoid confusions, we have revised the writing as “The phenomena of oncogene addiction and tumor suppressor gene hypersensitivity suggest that the multistage process of carcinogenesis, consistent with the fact that cancer-related proteins affect interacting networks, is not simply a summation of the isolated effects of oncogene activation and tumor suppressor gene inactivation”.

Point 3:“There is a general consensus for cancer genes that indicates that mutations in more than 1% of genes contribute to human cancer” 

Response 3We have revised the sentence as “There is a general consensus for cancer genes; mutations in more than 1% of genes contribute to human cancer”.

Inappropriate expression:

Point 4:“As is well known, mutations play an important role in cell proliferation but also in cell fate determination, such as differentiation”. 

Mutation has multiple effects on cell proliferation, or differentiation, or cell fate determination, but one should not say mutations play important role in those processes. 

Response 4Sorry, we misused the expression. We revised this statement as “As is well known, mutations have multiple effects on cell proliferation but also in cell fate determination, such as differentiation, apoptosis, and senescence”.

Point 5:“These studies accelerated the quest for considerable complexity in the mutational origins of cancer, with cancer- causing genes varying across and within tumor types and with multiple genes contributing to carcinogenesis”

What does “These studies” refer to in the context of this paragraph?

Response 5Sorry for description missing. We have revised “These studies” as “The researches on oncogenes and tumor suppressor genes” in order to make it more descriptive.

Point 6:Some sentence should not appear on such a professional review. For example:

 “Genes are coded messages that produce proteins, and the proteins control the way cells behave.”

“The HGP, nominally completed in 2004, aimed to sequence the human genome [34]. The

genome-wide analysis has provided substantial information for understanding genes and their mutations; reports of genome sequences ripple beyond biomedical science and technology into the social, economic, and political spheres.”

Response 6Sorry, these statements were the excessive expression in a professional review. We deleted all of the inappropriate expressions.

Reviewer 2 Report

The manuscript by Shimomora et. al. describes what is synthetic lethality, it’s history and application in efforts to find a lung cancer cure. The authors did a very nice job in explaining the concepts, disease and applications. The manuscript is well organized and I truly enjoyed reading it.

The topic of synthetic lethality is hot these days, in particular in the field of lung cancer. I have no concerns regarding this manuscript.

Author Response

Response to Reviewer 2 Comments

We are grateful to the reviewer for their critical comments and suggestions that have helped us to improve our manuscripts. 

Reviewer 2:

The manuscript by Shimomura et. al. describes what is synthetic lethality, it’s history and application in efforts to find a lung cancer cure. The authors did a very nice job in explaining the concepts, disease and applications. The manuscript is well organized and I truly enjoyed reading it.

The topic of synthetic lethality is hot these days, in particular in the field of lung cancer. I have no concerns regarding this manuscript.

Response: Thank you for your comments. We will keep studying.

Round  2

Reviewer 1 Report

no specific comments.